# The Technology Acceptance Model and Older Adults’ Exercise Intentions—A Systematic Literature Review

**DOI:** 10.3390/geriatrics7060124

**Published:** 2022-11-02

**Authors:** Yi Yau, Chia-Huei Hsiao

**Affiliations:** 1Department of Physical Education and Sport Science, National Taiwan Normal University, 162, Section 1, Heping E. Rd., Taipei City 106, Taiwan; 2Department of Leisure and Sport Management, National Taipei University, 151, University Rd., San Shia District, New Taipei City 23741, Taiwan

**Keywords:** TAM, sportstech, VR technology, wearable device, exergame

## Abstract

Aging is a global phenomenon, and the use of exercise technology by older adults can help them to prevent disease, achieve good health, and ultimately achieve successful aging. In the past, there literature compilation studies have been conducted on sports technology and young people or on the use of technology by the older adults; however, no studies have determined the attitudes of older adults toward sports technology. This review applied a systematic literature analysis to determine the factors that correlate the technology acceptance model with the older population’s exercise attitudes. A total of 10 studies were identified as contributing to the use of exercise technology by older adults. The main findings of this review are that, of the 28 factors identified in the 10 studies, only 18 were identified as factors influencing older adults’ use of sports technology in the technology acceptance model (TAM). Among these, fifteen factors affected intention, four factors affected perceived ease of use, three factors affected perceived usefulness, and two factors affected attitudes. Finally, discussing the related factors affecting TAM allows us to provide suggestions for future research directions.

## 1. Introduction

According to the United Nations’ World Population Prospects report (2019), the older population has grown rapidly [1]. The number of people over 65 years old rose from 607,548,000 in 2015 to 727,606,000 in 2020, an increase of 16.5%, and it is estimated that the population over the age of 65 will rise to 16% in 2050, up from 9% in 2019. Therefore, it is important to focus on the health of older adults so as to achieve successful aging. Successful aging refers to the avoidance of disease and disability, the maintenance of high physical and cognitive functions, and sustained engagement in social and productive activities [2]. In a systematic review of the research on successful aging, it was pointed out that successful aging can be observed from the perspectives of five structures, namely physiology, well-being, engagement, personal resources, and extrinsic factors [3], and the most common significant factors associated with successful aging are age, non-smoking status, and the absence of disability, arthritis, and diabetes [4]. It can be seen that preventing disease and maintaining health are the keys to successful aging, and the most direct way of becoming healthy is to exercise. However, during the COVID-19 pandemic, older adults became less likely to go out and even lost the opportunity to visit parks, for example, for exercise. Therefore, sports technology can help them to exercise at home and improve their sports opportunities. Sports technology (sportstech) is a broad range of artificial means for realizing human interests and goals related to sports [5]. Much of this technology targets sports, including wearables [6,7], fitness apps [8], exergames [9,10,11,12,13], and VR sports [14,15]. In recent years, the use of sports and wellness technology has become more popular, as the variety of applications and devices has increased [16]. For example, IT technology is being applied more frequently to mitigate the increasing health demands of older adults [17]. Thus, technological advancements, in recent years, have created opportunities to develop applications that support training at home, especially for older adults, as they can be more socially isolated and less physically active, and they have fewer opportunities to receive training in fitness centers [18]. In their research, Campelo and Katz point out that new exercise technologies, such as exergames and wearables, can enable older adults to improve their physical fitness and stimulate their interest and sustained physical activity participation, adding more benefits than previous physical activity and motivational incentives [19]. It can be seen that technology plays an important role in the successful aging of human beings [20].

However, with the development of sports technology, most studies have focused on younger users, ignoring older users. Therefore, we require more research on the use of sports technology by older adults in order to identify ways to effectively promote its use. One of the most commonly used models for understanding technology use is the technology acceptance model (TAM), as it focuses on the practical aspects of technology. TAM has been applied in different fields, and technology adoption by TAM is largely determined by the intent to use a particular type of technology [21,22]. TAM is an extension of rational action theory [23]. Its elements include attitude, perceived usefulness, perceived ease of use, and intent. Perceived usefulness and perceived ease of use are key factors in TAM and affect intent. Other attitudes, including personal beliefs, are also important for intent [24]. The technology acceptance model was also identified as the most widely used model in prior studies and systematic literature reviews on the intent to engage with fitness and physical exercise applications [21], while another study confirmed the extension of TAM to wearable fitness technology and supported this model [24]. It can be seen that the technology acceptance model is widely used in sports technology research.

The study of Günthner pointed out that previous research showed different findings in regard to the technology acceptance model of the older adults [25]. The research of Gune and Acarturk compared the technology acceptance patterns of older adults and young people and found that perceived ease of use and usefulness have impacts; however, external variables have different effects on technology use and acceptance by older and younger citizens [26]. It can be seen from the abovementioned study that the factors influencing older adults and young people are different; thus, it is necessary to identify the influences of different variables on the TAM of sports technology for older adults in order to promote their sports intentions. Among the evolving technology acceptance models, the Davis [27] model of technology acceptance is the most well-proven, robust, and influential model for studying IT acceptance [25]. Therefore, this study uses the Davis [27] model of TAM as an audited version, with dimensions including perceived usefulness, perceived ease of use, attitudes, and intentions.

A previous study [28] conducted a systematic literature review on TAM in health informatics. The purpose of [28] was to research health. It was impossible to know the impact of this technological product on the exercise intentions of the research subjects, and the study conducted a systematic literature review based on all the research subjects. However, as previously mentioned by [26], young and old people may have different results in relation to TAM. Another previous study conducted a systematic literature review on the intention of the older adults to use technology [29], but the review lacked any literature analysis of sports technology for older adults. Thus, this study provides the relevant literature. Moreover, the review’s [29] results are relevant to older adults using different kinds of technology, and it is expected that the results of this study will pay greater attention to the use of exercise technology among older adults in an attempt to understand the factors affecting TAM and the more effective promotion of exercise intentions.

Although the technology acceptance model has been used for many years, the use of technology by older adults is still not common, and the adoption rate is very low [30]. Previous research on older technology devices indicated that older adults in Brazil [31] and the United Kingdom [32] expressed concerns about using technology because of the fear of damaging new and expensive gadgets, and they showed great resistance towards and fear of this technology. Therefore, understanding the factors that affect TAM in older adults can help to improve the acceptance of technology by older adults in the future. Technology is recognized as an effective means enabling older adults to remain healthy, independent, safe, and socially connected [30]. However, there are few studies focusing on the older population, and most of them focus on rehabilitation research, especially the use of sports technology in older adults. Intentions have less impact. Today, the use of technology by older adults has gradually improved, increasing their adaptation to the technological social environment and meeting their need to functionally adapt to an increasingly technological social environment. Hence, understanding this technological adaptation has become an essential part of social and gerontological research [33]. In the future, as older adults become increasingly knowledgeable about technology, the use of sports technology is expected to become a new method of performing sports among older adults. Investigations of the older population’s acceptance of sports technology may be useful in our efforts to develop products that are suitable for them, as well as older adults’ welfare technology, thereby enhancing their exercise intentions and actual exercise behaviors so as to achieve successful aging. With the gradual popularization of sports technology, the price of these products has lowered, meaning that the entire older population—whether on a high or low income—can use sports technology to exercise. In terms of sustainable development, the research on sports technology for older adults can not only promote the health and wellbeing of older adults but also help to reduce inequality and enable older adults to embrace sports by, for example, using virtual fitness trainers via mobile applications. While TAM has been identified as a factor influencing user adoption intent, external factors are still being discovered, and these external factors will continue to differ between older adults and the young [26].

Based on this knowledge, the main purpose of this study was to review the relevant factors affecting the TAM model with a focus on the use of sports technology among the older population through a systematic literature review. Some researchers have combined technology acceptance patterns with other factors. However, no researchers have summarized the important factors related to the use of sports technology by older adults. Therefore, this study determined these aspects, which can be used to understand the current research directions, and analyzed the important results as a reference for the relevant practitioners and future researchers, on the understanding that identifying the influences of the factors can lead to effective strategies for promoting sports-technology-related products for older adults.

## 2. Materials and Methods

A systematic literature review was conducted based on the recommendations of Tranfield et al. [34] and the general guidelines of the Preferred Reporting Items for Systematic Reviews and Meta-Analyses (PRISMA).

### 2.1. Search Strategy

Relevant studies were identified through an electronic search of the Scopus and EBSCO databases. Based on peer-reviewed articles published between January 2011 and December 2021, keyword combinations were selected to determine the relevant studies for this research. We ensured that the delimitation included all aspects of the research project. The following combination was selected as the final search phrase: “Technology Acceptance Model” AND “elderly” OR “older adults.” The search was restricted to peer-reviewed scientific journals (446 in Scopus, 302 in EBSCO) and resulted in a total of 748 papers (3 March 2022) (Figure 1).

### 2.2. Inclusion and Exclusion Criteria

Articles were included if they met two criteria: (a) a focus on a sports technology product; (b) a focus on exercise intentions. Articles were excluded if the technology was for general health purposes. For example, mHealth is used for health management, not for exercise purposes. A total of 748 articles underwent title and abstract review by 2 authors, of which 704 did not meet the inclusion criteria. A total of 44 articles were selected.

### 2.3. Study Selection

After that, 44 articles were exported to Excel, and duplicate articles appearing in 2 databases were excluded, which included a total of 8 articles. There were then 36 articles in total. Then, the eligibility assessment was carried out, removing review articles (*n* = 6), those that did not focus on older adults (*n* = 3), conference articles (*n* = 12), articles that were not in English (*n* = 1), and those not related to the technology acceptance model (*n* = 4), including a total of 26 articles. Finally, a total of 10 articles were included for further analysis and reviewed by two authors.

### 2.4. Data Extraction

Each article was coded according to its associated factors, contextual factors, and methodological approach [35]. The data included the journal title, publication year, author name, abstract, author keywords, study location, study design and method, study population, number of study subjects, recruitment method, study objectives, the environment for using sports technology, study variables, main results, and conclusions.

## 3. Results

### 3.1. Methodological Approach and Samples

First, the methodologies of the related studies were examined to determine the main research methods (Table 1). Most of the studies were based on the quantitative method of questionnaires, while other studies used mixed methods, with a combination of questionnaires and semi-structured interviews. Secondly, the total number of older adults included in the 10 studies was 448, with a minimum of 11 older adults and a maximum of 146 older adults. Hence, the average number of participants was 44.8 in these studies.

### 3.2. National Differences

Taiwan (40%) appeared to be the leader in research on senior sports technology, and one study was conducted in each of the following countries: India, Switzerland, China, Canada, the USA, and the United Kingdom (10% each) (Table 2). None of the studies included multiple countries.

### 3.3. Age Range of the Older Adults

A systematic review of the literature found that most national studies focused on older adults aged 60 or over. However, the targeted older populations in Canada and the United Kingdom were aged 55 and 59 or over, respectively (Table 3). Thus, different countries had different definitions of older adults.

### 3.4. Recruitment Methods

A systematic review of the literature found that the participants were divided into the following categories: general communities (*n* = 3) [6,11,36], which are residential communities without special age divisions; older adults communities (*n* = 3) [7,9,12], which are residential communities specifically designed for healthy older adults; and learning centers (*n* = 2), which can be subdivided into lifelong learning centers [14] with no age limit and senior learning centers [15] with age limitations. One of the studies focused on assistance centers [10], referring to centers assisting residents with mental or physical disabilities (Table 4). Four of the studies [7,9,11,12] targeted healthy older adults during the recruitment or exclusion criteria process, while others did not specify exclusion criteria for the sampling.

### 3.5. Sports Technology

According to the systematic review of the literature, the most common research subject in regard to sports technology was exergames, which addressed the technology acceptance level and attitudes of the older adult research subjects toward participating in an exergame (Table 5). The second most common form of sports technology was a wearable device, with studies addressing the technological acceptance level and attitudes of the older adult research subjects toward the use of wearable devices. This was further subdivided into wearable bands and wearable vests. Interestingly, there were two articles using immersive VR glasses with VR technology. It can be seen that the older population is becoming increasingly widely accepting of new technologies. Moreover, according to the systematic review of the literature, all existing sports technology is for personal use, and its use environments are mainly public places. The main reason for this is that the equipment is large and expensive, so that the study participants cannot use it at home.

Half of the articles in this review are based on exergames, representing motion technology. Therefore, we will further analyze the movements of exergames (Table 6). Exergames include a combination of exercise and gaming that uses interactive video games to change attitudes towards health, primarily by training the user’s balance while ensuring that physical activity results in high energy expenditure, which can promote healthy behaviors [11,13]. There are five studies related to exergames in this review, four of which include low-intensity exercise [9,10,13] and one of which focuses on high-intensity exercise [11]. In addition, the exergame review mainly focuses on lower extremity movements [9,10,11,13], and less research [9,12] focuses on upper extremity movements. In game design, muscular strength is the main focus, along with reaction times [9,12] and balance [10,12].

### 3.6. Key Constructs Used in the Reviewed Studies

According to the key constructs in the articles, there were a total of 28 key constructs in the research on sports technology and older adults in regard to the technology acceptance model (Table 7). Perceived ease of use (*n* = 9), perceived usefulness (*n* = 8), attitude (*n* = 4), and intention (*n* = 7) were among the more frequently studied technology acceptance models. In addition, the TAM usually included external factors. Among the studies in this systematic review, norms (*n* = 3), facilitating conditions (*n* = 2), and experience (*n* = 2) were considered.

### 3.7. The Relationship between the Technology Acceptance Model and the Use of Sports Technology by Older Adults

Table 8 and Table 9 show the main results regarding the technology acceptance model and the older population’s use of exercise technology (Table 8 and Table 9). Among the 10 studies in the analysis, there were 18 factors that affected the reasons for the older population’s use of exercise technology. In addition to the three major factors (attitude, perceived ease of use, and perceived usefulness) of the technology acceptance model, in terms of external factors, there were 15 factors that affected the use of sports technology by older adults, including compatibility, effort expectancy, experience, facilitating conditions, habit, health conditions, hedonistic motivation, image, norms, output quality, perceived enjoyment, perceived playfulness, performance expectancy, price value, and risk. The most important influencing factors were the effect of perceived usefulness on intention and the effect of perceived ease of use on perceived usefulness, which are the two factors that have been identified in many studies. Furthermore, among the external factors, facilitating conditions and perceived playfulness stood out in two studies and have been shown to have an effect on intention.

An interesting study reported in [12] pointed out that older adults’ willingness to use a sports platform for a second time improved with the increase in the perceived usefulness. The study of Yein and Pal identified the strongest significant positive relationship between hedonistic motivation and intention [13], as did the qualitative responses reported in [11]. These studies suggested that the enjoyment of using the exercise technology helped participants to forget that they were exercising. The respondents claimed that they intended to use sports technology in the future. Among the other emotions, perceived enjoyment and perceived playfulness also affected the older population’s intention to use sports technology.

In a mixed study, Rebsamen et al. showed that intentions and attitudes may determine the high levels of motivation to use exergames on a regular basis [11]. Meeks and Stanmore showed that older adults have a more positive attitude [10], and the importance of social interaction was one of the main findings of this study. Puri et al. stated that device characteristics play the most important role, such as the display, battery, comfort, and aesthetics [36]. In terms of other aspects of the equipment, the compatibility of the sports technology’s ontology had an impact on the use of said sports technology by the older adults. Compatibility affected the perceived ease of use, perceived usefulness, and intent. Compatibility refers to the degree to which technology and other existing products (such as smart phones and tablet) fit the needs and lifestyles of the users [6]. Facilitating conditions had an impact on the perceived ease of use and intention, including whether users had the knowledge necessary to use the technology, the finances to acquire it, and assistance from others when using it [6].

The 10 studies all pointed out that the older adults showed a positive acceptance of sports technology. Previous studies directly indicated that the scores for the perceived ease of use and perceived usefulness of sports technology among older adults were high, which shows that the sports technology was highly accepted [11,14].

**Table 8 geriatrics-07-00124-t008:** The influence of the technology acceptance model on the use of sports technology among older adults.

Factor	Affected Factor	Total
Attitude	Intention [9,12],	1
Perceived Ease of Use	Attitude [12], Perceived Usefulness [6,12,15], Intention [26]	3
Perceived Usefulness	Attitude [12], Behavior [9], Intention [6,9,12,15]	3

**Table 9 geriatrics-07-00124-t009:** The influence of the external factors on the use of sports technology among older adults.

Factor	Affected Factor	Total
Compatibility	Intention [6], Perceived Ease of Use [6], Perceived Usefulness [6]	3
Effort Expectancy	Intention [13]	1
Experience	Perceived Usefulness [15], Perceived Ease of Use [15]	2
Facilitating Conditions	Intention [6,13], Perceived Ease of Use [6]	2
Habit	Intention [13]	1
Health Conditions	Intention [6], Perceived Ease of Use [6]	2
Hedonic Motivation	Intention [13]	1
Image	Intention [9]	1
Norms	Intention [15], Perceived Usefulness [6], Behavior [9]	3
Output Quality	Perceived Playfulness [9]	1
Perceived Enjoyment	Intention [15]	1
Perceived Playfulness	Behavior [9], Intention [9,13]	2
Performance Expectancy	Intention [13]	1
Performance Risk	Perceived Usefulness [6]	1
Price Value	Intention [13]	1

Among the studies that provided evidence on the technology acceptance model and the use of sports technology by older adults, most attempted to identify the factors influencing the intention to use sports technology among older adults (Table 10). The studies of Cook and Winkler and Rebsame et al. pointed out that the scores for perceived ease of use and perceived usefulness of the older adults using sports technology were high, presenting a high acceptance of sports technology [11,14]. Among the 18 factors mentioned above, 15 factors had an impact on intentions, including attitude, perceived ease of use, perceived usefulness, compatibility, effort expectancy, facilitating conditions, habit, health conditions, hedonistic motivation, image, norms, perceived enjoyment, perceived playfulness, performance expectancy, and price value. There were four factors that affected perceived ease of use, namely compatibility, experience, facilitating conditions, and health conditions. There were three factors that affected perceived usefulness, including compatibility, experience, and norms. There were two factors that affected attitudes, namely perceived ease of use and perceived usefulness.

## 4. Discussion

The objective of this study was to conduct a systematic review to identify studies that have examined the technology acceptance model of older adults using sports technology. A total of 18 factors were presented as affecting the use of sports technology by older adults, including three factors in the technology acceptance model. To facilitate the discussion of these findings, we organized them into five areas: (1) the methods and samples used in the studies; (2) the study location, definition of advanced-age older adults in that study location, and the recruitment method; (3) the sports technology used in the studies; and (4) the influencing factors identified in the studies.

***Methods and samples used in the studies.*** According to the research methods of the 10 studies identified in the systematic literature review, 50% used questionnaires as the research method, and the other 50% used qualitative interviews combined with questionnaires as a mixed method. The qualitative interview method can help us to better understand the thought processes of older adults, because it may be difficult for them to fill in the questionnaire if they are illiterate or have an education level below primary school [6]. Educational levels may influence older adults’ use of technology [37], and a lack of investigation regarding educational levels in some studies may lead to potential bias [10,13,15]. Thus, through mixed method interviews, respondents can more fully explain the reasons for their technology acceptance patterns and the cognitive principles behind the questionnaire results [36]. Some mixed studies did not explain the interview steps or process [13,14]. Furthermore, information about the number of people who were involved in conducting the interview and processing the interview content was omitted, which affected the consistency of the results. Only two studies had more than 100 participants fill out the questionnaires [6,12], while the other studies were small. Some studies had fewer than 30 participants [13,36], and some studies had fewer than 20 participants [10,11,14]. Sample sizes that are too small cause biased sampling, but most of the studies were based on longitudinal research involving three to six weeks of exercise interventions. Therefore, the study samples were not large. In terms of the gender ratio of the sample, females were 1 or more times more prevalent than males [7,11,12,15]. The gender ratio should be as close as possible to avoid biased sampling, which leads to results biased towards one gender. Two studies [9,12] did not specify the timing, and one study [14] stated the number of weeks of the exercise intervention, but it did not specify the time of each exercise session. The time of the exercise intervention was inconsistent between study participants, which may cause potential bias.

***Study location, definition of older adults in the study location, and methods of recruitment.*** In total, 40% of the studies were conducted in Taiwan. This finding suggests that there is a clear geographic concentration in sports technology research on older adults’ technology acceptance models. Ho et al.’s research showed that Taiwan became an aging society in 2018, and it is expected that it will become a super-aging society in 2025 [38]. The aging population is growing rapidly, and health problems will also increase. However, the Taiwanese government has attached great importance to the health of older adults. It has promoted various aspects of sports policies and research for older adults, such as Sports I, Taiwan’s silver-haired sports program, and has even established sports centers for older adults and provided technology-based sports equipment. These approaches can promote sports technology for older adults. Interestingly, all 10 studies were studies on older adults, with the average age of the sample being around 60–70 years old. However, the minimum age of the participants in each study was different. In particular, the youngest participant in Puri et al.’s research was 55 years old [36], while the youngest participant in the research of Meekes and Stanmore was 59 years old [10]. These two studies differed from the UN’s definition of older adults as being over the age of 60 [39]. For the recruitment process, some studies did not explain the process of recruitment [13] or only presented the recruitment area [15]. The process of selecting the sample can affect the representativeness of the results and cause potential bias. Some exergame studies [13,14,15] did not explain the exclusion of physical conditions at the time of recruitment; thus, we did not know whether there was consistency in the physical conditions of each of the study participants, which affects the representativeness of the sample and causes sample bias. Most studies did not state whether the people who used the product in the study were excluded from the recruitment [6,7,9,10,12,13,14,15,36]. Experienced and non-experienced study participants, who were at different baselines, may have affected the results.

***Sports technology used in the studies*.** The types of sports technology used in these studies can be divided into three types: exergames accounted for 50%, wearable devices accounted for 30%, and VR technology accounted for 20%. Piech and Czernick defined exergames as the activity of playing video games involving physical exertion, and interventions utilizing exercise games had favorable effects on both motor and cognitive functioning [40]. This is a form of intervention that improves physical functioning in older adults, with few adverse effects. Zheng et al. reviewed the effects of exergames on the physical condition of frail older adults through a systematic literature review and concluded that exergames showed a tendency to increase muscle strength when combined with resistance training. In terms of equipment, exergames are mainly equipped with Kinect sensors that monitor the user’s physical movement angles [41]. Sato et al.’s research showed that Kinect provides the unique possibility of quantifying the individual’s balance ability when performing complex tasks in a sports game environment and provides an opportunity for at-home training and the assessment of postural control among the older population, but the technical and cost issues of the sensor need to be resolved [42]. Regarding the rapid development of sensors, Yu et al. found that movements performed during physical activity can be assessed and corrected in real time through a single camera, and the human body’s key points can be used to monitor the user’s movement [43]. As for the use of sports technology by older adults, the users are individuals, and there is a lack of team sports. Previous research has pointed out that group sports technology can improve older adults’ physical literacy and willingness to use it [19]. In terms of the use environment, sports technology is mainly used by the older adults in public places. Meeks and Stanmore showed that the use of exergames in public spaces can promote communication with others, and social interaction encourages the older adults to play exergames [10]. In the future, the convenience of using the technology at home should be considered, and the flexibility of its use should be improved. In addition, from the perspective of exergames, sports technology by older adults is mainly designed for low-intensity exergames for older adults in terms of the organization. Only one study included mainly high-intensity interval training, and there is a lack of exercise intensity choices for healthy older adults. Previous studies on the use of high- and moderate-intensity interval training in nursing homes among older adults indicated that HIIT had greater benefits for the body composition and functional performance [44]. It can be seen that even older adults living in assisted living facilities can engage in higher-intensity exergames.

***Influencing factors identified in the studies.*** Although there were a total of 28 factors in the 10 studies, only 18 factors were identified as influencing older adults’ use of sports technology in the technology acceptance model. Among these, fifteen factors affected intention, four factors affected perceived ease of use, and three factors affected perceived usefulness, while among attitudes, only the perceived ease of use and perceived usefulness of the TAM model had an effect. This is consistent with the findings of Chen and Chan on the use of geriatric technology by older adults, in that the direct influence of attitude-related factors on the use of geriatric technology is not significant [45]. Previous studies [26,27,45] have identified relationships between attitudes, perceived ease of use, perceived usefulness, and intention, with most of them focusing on the influences of other factors on intention, ignoring the remaining three factors. In particular, there is a close relationship between the perceived ease of use and perceived usefulness; thus, this study mainly explored the impacts of external factors on these two factors.

The device compatibility and experience were important in terms of both the perceived ease of use and perceived usefulness. In terms of the device compatibility, Frennert et al. found that older adults prefer fewer electronic devices and better-integrated functions [46], and low energy consumption is an essential characteristic of such wearable and battery-operated monitoring devices [47]. However, some respondents stated that self-monitoring devices caused additional stress and anxiety [46]. The reasons for the use of technology by older adults include the fear of failure, as mentioned above [31,32], which may be due to security and privacy [47,48]. Therefore, it is necessary to consider the pressure that an older adult may feel when using a device, which may be due to the lack of experience of older adults in using electronic products. Personal experience in using electronic products was also very important. Therefore, in the future, pre-use teaching should be provided to strengthen people’s experience with electronic products and reduce the pressure they may feel. However, while previous experience with technology may influence its use in older adults [37], some studies [7,10,12,13,15] did not consider past experience with technology, which can lead to potential bias.

In terms of the perceived ease of use, the facilitating conditions of a device were important. The study of Puri et al. directly pointed out that the display, battery life, comfort, and aesthetics of wearable wristbands were valued by older adults [36], and Chen and Chan determined that facilitating knowledge, guidance, support from others, and the level of accessibility allowed older adults to use the technology [45]. Stamm’s research directly points to the importance of technical support for older adults in regard to exercise using sport technology [48]. Oppl and Stary’s study found that explanations, demonstrations, coaching, or the opportunity to ask questions encouraged progress after older adults experienced problems using technology, as well as the interesting finding that older adults preferred to seek help from a counselor rather than a partner [49]. Although older adults do not wish to seek help from their partners, norms do have an impact on perceived usefulness, and older adults perceive the opinions of significant others to influence their perceptions of the usefulness of sports technology products. Galli et al. showed that older adults’ perceptions of significant others could create an environment supportive of their autonomy, which not only fosters their autonomous motivation but also positively correlates with behavioral beliefs [50].

Health conditions had an impact on the perceived ease of use, which is usually observed in virtual sports, as most older adults feel that their health conditions are poor and that they lack the ability to engage in sports. Furthermore, they believe that virtual interfacing sports are more difficult than traditional sports. Therefore, a physical fitness test can be carried out first to enhance their confidence in their own health conditions. The advantage of virtual sports is that they provide the opportunity to engage in sports that are difficult to implement in reality [14], which can improve the older adults’ confidence in engaging in other sports.

Finally, perceived playfulness had an impact on the older adults’ intention to use exercise, and the same finding was observed in Damberg’s research on the use of fitness apps by adults in general [8]. The research showed that users of technology perceive the impact of playfulness as important, and users tend to increase their use of technology when it is perceived to have entertaining and creative functions. Therefore, the development of sports technology for older adults should render the product design more interesting, enable the older adults to be immersed in the game, and reduce the perceived difficulty when exercising.

## 5. Conclusions

According to the results of this review, the sample sizes of the previous studies were too small. Therefore, the sample size should be increased, and the age of all the respondents should be strictly controlled so as to meet the scope of the study. In the research process, we should pay attention to these matters to avoid potential bias, including the sample size, gender ratio, sample exclusion criteria (physical conditions, accounting for whether the participants have used the product before), interview process, consistency in the exercise intervention time, technology use experience, and education level. In the product development process, product integration functions should be added in the future, and sports games should be made interesting in order to enhance the enjoyment of the older adults so as to increase the rate of reuse. Before using sports technology products, health conditions may cause the older adults to have doubts about their competence; hence, physical fitness tests can be provided to help the older adults to improve their self-confidence in their own health. In addition, under the influences of the past experience of the use of electronics and convenient conditions, it is recommended that, in the future, the relevant personnel should enable older adults using sports technology to engage with counselors, and counselors should provide pre-use instruction to enhance the experience of older adults, as well as coaching to reduce the stress experienced when using the technology, in addition to increasing the choice of exergames, including HIIT in the exercise program, so that healthy seniors have the options of different exercise intensities. Moreover, exergames are currently used in public places, and the convenience of using them at home should be considered. At present, the self-monitoring system of mobile phones is well-developed, and these should be considered as alternative sensing devices to Kinect so that older adults may also engage in exergames at home and improve their sports opportunities.

In terms of academic research, the focus has remained on the effect of sports technology on physical fitness; however, an understanding of the complete technology acceptance model is lacking. In the future, researchers should focus on the feelings of older adults and add qualitative interviews to questionnaires in order to understand the reasons behind their answers to the questionnaires. In the abovementioned research, wearable devices and virtual sports were independently examined. With the advances in science and technology, sports-technology-based wearable devices have been developed. If exergames and wearable devices were to be combined, this would allow researchers to more effectively and accurately comprehend the physical condition of older adults. Under this type of monitoring system, the older adults may be affected by increased risks and anxiety; thus, researchers should study these two aspects in the future. In the above discussion, we observed that team sports can improve the physical literacy of older adults, enhance social interaction, and increase the intention to use sports technology. The current exergame studies lack research on the impact of group sports. In the future, researchers should focus on the impact of group exergames on older adults.

## Figures and Tables

**Figure 1 geriatrics-07-00124-f001:**
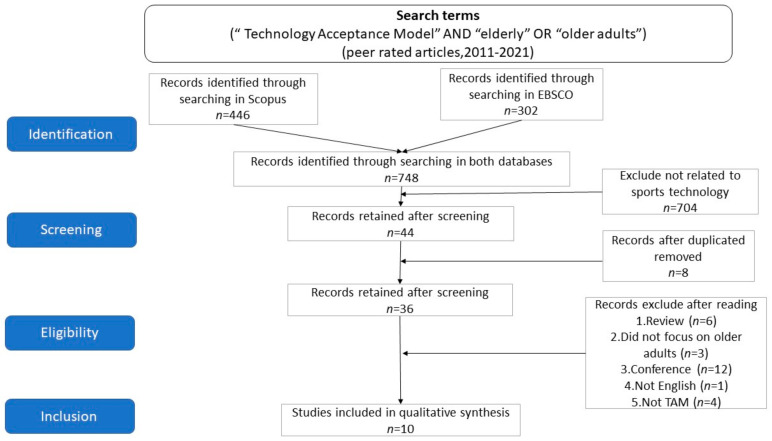
Preferred Reporting Items for Systematic Reviews and Meta-Analyses (PRISMA).

**Table 1 geriatrics-07-00124-t001:** Method and number of participants in the studies.

Method	Study	Number of Participants	Male	Female
Quantitative	[6]	146	82	64
	[7]	50	10	40
	[9]	39	15	24
	[12]	101	27	74
	[15]	30	6	24
Mixed Method	[10]	12	6	6
	[11]	12	2	10
	[13]	27	17	10
	[14]	11	4	7
	[36]	20	8	12

**Table 2 geriatrics-07-00124-t002:** Number of studies per country.

Country/Region	Total Number of Articles	%
Canada	1 [36]	10%
China	1 [21]	10%
Switzerland	1 [11]	10%
Taiwan	4 [7,9,12,15]	40%
India	1 [13]	10%
USA	1 [14]	10%
United Kingdom	1 [10]	10%

**Table 3 geriatrics-07-00124-t003:** Minimum age of the older adult participants in the studies.

Country/Region	Study	Minimum Age of Participants
Canada	[36]	55
China	[6]	60
Switzerland	[11]	65
Taiwan	[9]	62
	[7]	60
	[15]	60
	[12]	60
India	[13]	60
USA	[14]	71
United Kingdom	[10]	59

**Table 4 geriatrics-07-00124-t004:** Recruitment in the studies.

Area	Study	Recruitment
No description	[13]	No description.
General Community	[11]	Recruited through local newspaper advertisements, bulletins and billboards in local stores, and/or through the personal contacts of research staff. Study participants must be untrained and have self-reported good health.
	[36]	Flyers posted through local community centers and recreational facilities, using convenience sampling.
General community and Park	[6]	Recruited through local residential communities and parks, using convenience sampling.
Learning Centre	[14]	Recruited through local lifelong learning academies; study participants are active and educated older adults in nearby communities.
	[15]	Recruited through a local ageing learning center.
Older adults Community	[12]	Recruited through local senior care communities, where seniors can live independently without physical or cognitive impairments.
	[7]	Recruited through local senior care communities, where seniors can live independently without physical or cognitive impairments.
	[9]	Recruited through local senior care communities, where seniors can live independently without physical or cognitive impairments.
Assisted Living Center	[10]	Recruited seniors living in local assisted living facilities, where the occupants have a disability (mental or physical) and the center has specialist staff to supervise.

**Table 5 geriatrics-07-00124-t005:** Different sports technologies of the studies.

Study	Sports Technology	Equipment	Use Environment
[9]	Exergame	Kinect	Public Places
[10]	Exergame	Kinect	Public Places
[11]	Exergame	Dividat Senso Smart	Public Places
[12]	Exergame	Wireless Remote Sensors	Public Places
[13]	Exergame	Capacitive Sensor	Public Places
[14]	VR Technology	Immersive VR Glasses	Public Places
[15]	VR Technology	Immersive VR Glasses	Public Places
[6]	Wearable Device	Wearable Band	Anywhere
[7]	Wearable Device	Wearable Vest	Anywhere
[36]	Wearable Device	Wearable Band	Anywhere

**Table 6 geriatrics-07-00124-t006:** Different exergames in the studies.

Study	Activities Part	Main Training	The Movement of Exergames
[9]	Upper Extremity	Flexibility	A living memory interactive wall is used to choose and recall life memories by moving the arm, combining physical activity and cognition.
[12]	Upper Extremity	Grip Strength	Grip test.
[13]	Lower Extremity	Muscular Strength	Standing with eyes open, walking straight and side-stepping.
[10]	Lower Extremity	Muscular Strength	Without lifting the heels during the exercise, one drives the submarine using squats and discovers treasures while avoiding threats.
[12]	Lower Extremity	Balance	Balance test.
[11]	Lower Extremity	Muscular Strength	Rocket exergame in HIIT mode for up to 2 min. As the player steps faster, the rocket accelerates faster.
[9]	Lower Extremity	Reaction	In the Interactive Floor Kick and Play game, players make movements in response to sudden changes in shapes and image patterns projected onto the floor.
[9]	Lower Extremity	Reaction	Players need to respond to emergencies on the touch screen on the wall, improving hand-eye coordination, and slopes are added to the gameplay to increase physical energy consumption.
[10]	Lower Extremity	Balance	Players move the bees with their knees bent to collect nectar and bring it safely to the hive.
[10]	Lower Extremity	Balance	To simulate participation in an auction, players must stand (with their arms crossed over their chest) and sit down at a specified time to bid on the auctioned items.
[10]	Lower Extremity	Balance	Players need to stand with their legs together and knees slightly bent, grab the designated item before it disappears, and move the designated leg to control the body in the process.
[10]	Whole body	Muscular strength	Players pick up items and carefully place them on the shelf and reach out to pick up and move objects using the game’s designated arms, avoiding dropping them.
[12]	Whole body	Flexibility	Flexibility test.
[12]	Whole body	React	Reaction time test.

**Table 7 geriatrics-07-00124-t007:** Key constructs used in the studies.

Key Constructs	Study	*n*	%
Perceived Ease of Use	[6,7,9,10,11,12,14,15,36]	9	16%
Perceived Usefulness	[6,7,9,11,12,14,15,36]	8	14%
Intention	[6,7,9,11,12,13,15]	7	13%
Attitude	[7,10,11,12]	4	7%
Norms	[9,15,36]	3	5%
Experience	[9,15]	2	4%
Facilitating Conditions	[13,36]	2	4%
Social Influence	[6]	1	2%
Compatibility	[6]	1	2%
Effort Expectancy	[13]	1	2%
Equipment Characteristics	[36]	1	2%
Habit	[13]	1	2%
Health Conditions	[6]	1	2%
Hedonic Motivation	[13]	1	2%
Job Relevance	[9]	1	2%
Output Quality	[9]	1	2%
Perceived Playfulness	[9]	1	2%
Perceived Enjoyment	[15]	1	2%
Perceived Social Risks	[6]	1	2%
Perceived Risks	[36]	1	2%
Performance Expectancy	[13]	1	2%
Performance Risk	[6]	1	2%
Price Value	[13]	1	2%
Privacy Concerns	[36]	1	2%
Image	[9]	1	2%
Technology Anxiety	[7]	1	2%
Usage Behavior	[9]	1	2%
Voluntariness	[9]	1	2%
Total		56	100%

**Table 10 geriatrics-07-00124-t010:** Different factors influencing the technology acceptance model.

Factor	Affected Factor	Total
Perceived Ease of Use [12], Perceived Usefulness [12]	Attitude	2
Compatibility [6], Experience [15], and Norms [6]	Perceived Usefulness	3
Compatibility [6], Experience [15], Facilitating Conditions [6], and Health Conditions [6]	Perceived Ease of Use	4
Compatibility [6], Effort Expectancy [13], Facilitating Conditions [6,13], Habit [13], Health Conditions [6], Hedonic Motivation [13], Image [9], Norms [15], Perceived Enjoyment [15], Perceived Playfulness [9,13], Performance Expectancy [13], Price Value [13], Attitude [9,12], Perceived Ease of Use [26], and Perceived Usefulness [6,9,12,15]	Intention	15

## Data Availability

Not applicable.

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
