# Peer review of "The Technology Acceptance Model and Older Adults’ Exercise Intentions—A Systematic Literature Review"

_geriatrics, 2022, doi:10.3390/geriatrics7060124_

Round 1
Reviewer 1 Report (Previous Reviewer 1)
The authors have done a great job revising the manuscript. The analysis of the different papers in the review now goes further and with the new tables included the manuscript gives an interesting picture of the state of the art in exercise games and the acceptance of these among older individuals.
Author Response
Thank you for your nice comments on our review. On behalf of all the contributing authors, we sincerely appreciate all your valuable comments and suggestions, which helped us in improving the quality of the manuscript.
Reviewer 2 Report (Previous Reviewer 2)
No further comments
Author Response
On behalf of all the contributing authors, we sincerely appreciate all your valuable comments and suggestions, which helped us in improving the quality of the manuscript.
This manuscript is a resubmission of an earlier submission. The following is a list of the peer review reports and author responses from that submission.
Round 1
Reviewer 1 Report
The study is a systematic review aimed to compile the current knowledge of how older individuals' acceptance of sports technology.
* Major comments
1. With only 10 studies forming the material for the analysis, the manuscript should provide a more fully explanation of the technology investigated in each study. The section in the Discussion on "Sports technology used on the studies" is mostly a discussion of what exergames are, and do not refer to the particular studies included in the analysis.
2. The analysis of the 10 studies should include how subjects were recruited, since this greatly influences how representative the samples are.
* Minor comments
1. Abstract and introduction: What does "successful" actually mean in this context? I suggest to be more specific.
2. Page 2: Make sure to consequently write TAM or tam, but not to mix.
3. Page 2 second paragraph: "perceived ease of use" repeated.
4. Page 2 fourth paragraph: "[...] most people think that the elderly are not suitable for technology". This sweeping statement doesn't seem accurate to me.
5. Page 3 Section 2.2: "Articles are exluded for health purposes". This seems too vague as an exclusion criterion. Please be more specific.
6. Page 3 Section 2.3: The 6 review articles that were excluded, although they should not be part of the analysis done in the systematic review, are probably relevant and should be related to in the discussion.
7. Page 4 Section 3.2: "There was little difference in reseach intensity [...]" The statement appears redundant, and not accurate, as there are many countries in the world not represented in the material.
8. Page 6 Section 3.5: Please provide a full reference for the VOSviewer software.
9. Page 7 Figure 2: There is way too much unused white space in the figure. The text is too small. The figure caption should give a much more fully explanation of the figure, in particular the significance of the colors.
10. Page 9 Figure 4: "Hedonistic motivation" should be "Hedonic motivation".
11. Page 11 second paragraph: Missing a motivation (reference) for why high-intensity interval training is particularly interesting.
12. Page 11 third paragraph: "Previous studies [...]". Missing references to the previous studies mentioned.
Author Response
Point 1: With only 10 studies forming the material for the analysis, the manuscript should provide a more fully explanation of the technology investigated in each study. The section in the Discussion on "Sports technology used on the studies" is mostly a discussion of what exergames are, and do not refer to the particular studies included in the analysis.
Response 1: This study has supplemented the analysis of exergames in sports technology.
Point 2: The analysis of the 10 studies should include how subjects were recruited, since this greatly influences how representative the samples are.
Response 2: This study has supplemented the relevant information on how to recruit.
Point 3: Abstract and introduction: What does "successful" actually mean in this context? I suggest to be more specific.
Response 3: This study has supplemented the specific meaning of successful aging.
Point 4: Page 2: Make sure to consequently write TAM or tam, but not to mix.
Response 4: This study has revised the full text to use TAM uniformly.
Point 5: Page 2 second paragraph: "perceived ease of use" repeated.
Response 5: This study has corrected the duplicated perceived ease of use to perceived usefulness.
Point 6: Page 2 fourth paragraph: "[...] most people think that the elderly are not suitable for technology". This sweeping statement doesn't seem accurate to me.
Response 6: This study has been corrected to describe the use of technology by older adults is still not a common phenomenon.
Point 7: Page 3 Section 2.2: "Articles are exluded for health purposes". This seems too vague as an exclusion criterion. Please be more specific.
Response 7: This study has added "for example, mhealth is used for health management not for exercise purposes".
Point 8: Page 3 Section 2.3: The 6 review articles that were excluded, although they should not be part of the analysis done in the systematic review, are probably relevant and should be related to in the discussion.
Response 8: This study has added some excluded articles into the discussion.
Point 9: Page 4 Section 3.2: "There was little difference in reseach intensity [...]" The statement appears redundant, and not accurate, as there are many countries in the world not represented in the material.
Response 9: The study has corrected this sentence.
Point 10: Page 6 Section 3.5: Please provide a full reference for the VOSviewer software.
Response 10: We have deleted this section considering that 10 studies were not effective enough for co-occurrence analysis.
Point 11: Page 7 Figure 2: There is way too much unused white space in the figure. The text is too small. The figure caption should give a much more fully explanation of the figure, in particular the significance of the colors.
Response 11: We have deleted this section considering that 10 studies were not effective enough for co-occurrence analysis.
Point 12: Page 9 Figure 4: "Hedonistic motivation" should be "Hedonic motivation".
Response 12: Figure4 "Hedonistic motivation"has corrected to "Hedonic motivation"
Point 13: Page 11 second paragraph: Missing a motivation (reference) for why high-intensity interval training is particularly interesting.
Response 13: We have deleted this section considering that 10 studies were not effective enough for co-occurrence analysis.
Point 14: Page 11 third paragraph: "Previous studies [...]". Missing references to the previous studies mentioned.
Response 14: This study has supplemented relevant references.
Reviewer 2 Report
1. The abstract is not informative. At a minimum, the authors need to state the main findings.
2. A native English speaker needs to check the manuscript for grammar and style. Such sentences lack scientific rigor and precision ‘…is a phenomenon that cannot be ignored…’; ‘…we should pay attention to….’; ‘…Generally speaking, the most…’ ‘…Refs. [10] pointed out that…’ < ---- this is not the way scientific referencing occurs.
3. The authors make a generalized statement as to how technology plays an important role in successful aging. This is however not a universal and even phenomenon across countries because, for example, in Eastern European countries there is a huge resistance against and even fear of technology among older people. As a result, there are many such countries where older adults are actually NOT receptive to IT technology in general and sports technology specifically. Similarly, in Brazil, for example, ‘…Of the elderly respondents, 24% reported being afraid to use new technological devices and 43.4% reported fear related to damaging the devices. The elderly responded that they feel afraid when using the internet; they fear viruses, social networks and spoiling or breaking the device.’ (Raymundo et al Fear and the use of technological devices by older people, DOI:10.4017/gt.2014.13.02.191.00). Also: ‘Older Adults Admit High Anxiety and Fear About New Technology: Candoo Tech Surveys Shows 53% of Seniors Say Learning a New Device is More Stressful Than Going to the Dentist’. And many more reports. Thus, the introduction needs to set the stage in a much more balanced manner.
4. The Introduction needs to have an element about general technology acceptance and then a specific form of it, sports technology.
5. I am not understanding the aim of this review. It states ‘….to review the relevant external factors that influence of technology acceptance models focusing’. The authors plan to examine the factors that affect the MODEL not older adults as to how they accept or not sports technology? I cannot follow this logic.
6. It is unclear from the paper whether or not the authors considered group use acceptance or individual use acceptance in a gym and / or home setting. Such critical factors are not described.
7. I cannot understand how the co-occurrence analysis is valid for such low number of studies. The authors must describe the validity of this method for such low frequency, as they base their conclusions on potentially flawed analyses.
8. Any conclusion based on n=1 study in Table 5 is unsubstantiated.
Author Response
Response to Reviewer 2 Comments
Point 1: The abstract is not informative. At a minimum, the authors need to state the main findings.
Response 1: This study has supplemented the main findings in the abstract.
Point 2: A native English speaker needs to check the manuscript for grammar and style. Such sentences lack scientific rigor and precision ‘…is a phenomenon that cannot be ignored…’; ‘…we should pay attention to….’; ‘…Generally speaking, the most…’ ‘…Refs. [10] pointed out that…’ < ---- this is not the way scientific referencing occurs.
Response 2: The first edition of this study has paid MDPI English editors to check the grammar and style of the manuscript. In order to ensure the English grammar and style of the revised version, the research has been paid again to the MDPI English editor and a grammar check of the manuscript has been completed.Additionally, this study has corrected sentences that lack science.
Point 3: The authors make a generalized statement as to how technology plays an important role in successful aging. This is however not a universal and even phenomenon across countries because, for example, in Eastern European countries there is a huge resistance against and even fear of technology among older people. As a result, there are many such countries where older adults are actually NOT receptive to IT technology in general and sports technology specifically. Similarly, in Brazil, for example, ‘…Of the elderly respondents, 24% reported being afraid to use new technological devices and 43.4% reported fear related to damaging the devices. The elderly responded that they feel afraid when using the internet; they fear viruses, social networks and spoiling or breaking the device.’ (Raymundo et al Fear and the use of technological devices by older people, DOI:10.4017/gt.2014.13.02.191.00). Also: ‘Older Adults Admit High Anxiety and Fear About New Technology: Candoo Tech Surveys Shows 53% of Seniors Say Learning a New Device is More Stressful Than Going to the Dentist’. And many more reports. Thus, the introduction needs to set the stage in a much more balanced manner.
Response 3: This study has supplemented relevant literature, and the literature provided by the reviewers has been included in the manuscript.
Point 4: The Introduction needs to have an element about general technology acceptance and then a specific form of it, sports technology.
Response 4: This study has supplemented the elements of general technology acceptance and the specific form of sports technology.
Point 5: I am not understanding the aim of this review. It states ‘….to review the relevant external factors that influence of technology acceptance models focusing’. The authors plan to examine the factors that affect the MODEL not older adults as to how they accept or not sports technology? I cannot follow this logic.
Response 5: This study has been revised for the purpose of writing.
Point 6: It is unclear from the paper whether or not the authors considered group use acceptance or individual use acceptance in a gym and / or home setting. Such critical factors are not described.
Response 6: This study has supplemented the description and discussion of the context of use and use by individuals or groups.
Point 7: I cannot understand how the co-occurrence analysis is valid for such low number of studies. The authors must describe the validity of this method for such low frequency, as they base their conclusions on potentially flawed analyses.
Response 7: We have deleted this section considering that 10 studies were not effective enough for co-occurrence analysis.
Point 8: Any conclusion based on n=1 study in Table 5 is unsubstantiated.
Response 8: This study has supplemented relevant literature in the table.